# Influence of Oxygen Flow Rate on the Properties of FeO$_X$N$_Y$ Films Obtained by Magnetron Sputtering at High Nitrogen Pressure

**Moussa Grafoute** [1,*], **Kouamé Boko Joël-Igor N'Djoré** [1], **Carine Petitjean** [2], **Jean François Pierson** [2] **and Christophe Rousselot** [3,*]

1   Laboratoire de Technologie, Université Félix Houphouët Boigny, UFR SSMT 22, Abidjan BP 258, Côte d'Ivoire; joelndjore@outlook.fr
2   Institut Jean Lamour—UMR 7198, Université de Loraine, 54011 Nancy, France; carine.petitjean@univ-lorraine.fr (C.P.); jean-francois.pierson@univ-lorraine.fr (J.F.P.)
3   Département MN2S, Institut FEMTO-ST (CNRS/UFC/ENSMM/UTBM), Université de Franche Comté, 25211 Montbéliard, France
*   Correspondence: gramouss@hotmail.com (M.G.); christophe.rousselot@univ-fcomte.fr (C.R.)

**Abstract:** Fe-O-N films were successfully deposited by magnetron sputtering of an iron target in Ar-N$_2$-O$_2$ reactive mixtures at high nitrogen partial pressure 1.11 Pa (Q(N$_2$) = 8 sccm) using a constant flow rate of argon and an oxygen flow rate Q(O$_2$) varying from 0 to 1.6 sccm. The chemical composition and the structural and microstructural nature of these films were characterized using Rutherford Backscattering Spectrometry, X-ray diffraction, and Conversion Electron Mössbauer Spectrometry, respectively. The results showed that the films deposited without oxygen are composed of a single phase of γ″-FeN, whereas the other films do not consist of pure oxides but oxidelike oxynitrides. With higher oxygen content, the films are well-crystallized in the α-Fe$_2$O$_3$ structure. At intermediate oxygen flow rate, the films are rather poorly crystallized and can be described as a mixture of oxide γ-Fe$_2$O$_3$/Fe$_3$O$_4$. In addition, the electrical behavior of the films evolved from a metallic one to a semiconductor one, which is in total agreement with other investigations. Comparatively to a previous study carried out at low nitrogen partial pressure (0.25 Pa), this behavior of films prepared at higher nitrogen partial pressure (1.11 Pa) could be caused by a catalytic effect of nitrogen on the crystallization of the hematite structure.

**Keywords:** reactive sputtering; Fe-O-N films; X-ray diffraction; Mössbauer spectrometry; electrical and optical properties

## 1. Introduction

Nowadays, the search for new materials with tunable physical properties is one of the greatest preoccupations of the scientific community. Several processes are well-established to synthesize new and/or metastable materials. Among them, the reactive sputtering process remains one of the most powerful for the deposition of high-performance thin films. Indeed, this process is widely used to deposit materials with a complex microstructure (nanocomposites, multilayers, etc.) [1], to synthesize new or metastable compounds [2] or to extend the solubility limit in binary or ternary systems [3].

In the past decades, iron oxide coatings have been extensively studied due to their applications in many industrial domains [4,5]. Since the development of transition metal oxynitride films [6,7], it appears that these compounds exhibit properties that can be tuned between those of nitrides and oxides by adjusting the nitrogen and the oxygen concentrations. Iron oxide films (Fe-O) are studied because of their optical and magnetic properties [8]. Iron nitride (Fe-N) films are studied because of their interesting magnetic properties [9] and their resistance to corrosion and wear [10]. The interest in the study of

iron oxynitride films (Fe-O-N) is to elaborate a material that exhibits intermediate properties between those of iron oxide and iron nitride films. The Fe-O-N thin films have been studied for potential application to photoelectrochemical water splitting to generate hydrogen, solar application [11], magnetic [12], optical, electronic, etc. properties [13,14].

At low nitrogen (or oxygen) content, this element could be dissolved into the oxide (or nitride) network. Since Mössbauer spectrometry is a relevant method to characterize the local environment of iron atoms, the study of Fe-O-N films could bring relevant information about the structure of transition metal (oxynitrides). However, little information is available in the literature on the structure and the properties of iron oxynitrides films [15]. M. Grafoute et al. [16,17] have prepared Fe-O-N films by magnetron sputtering of an iron target in Ar-$N_2$-$O_2$ reactive mixtures at low nitrogen partial pressure (0.25 Pa) using a nitrogen flow of $Q(N_2)$ = 2 sccm. However, the effect of the increase in nitrogen partial pressure or nitrogen flow rate on structural, chemical, and physical properties is not widespread in the literature.

In this work, we aim to study the influence of high nitrogen partial pressure (1.11 Pa) corresponding to $Q(N_2)$ = 8 sccm on the properties of Fe-O-N coatings formed by reactive magnetron sputtering of an iron target in Ar-$O_2$-$N_2$ reactive mixtures. In addition, the correlation between structural and physicochemical properties of Fe-O-N films will be discussed. Finally, we will compare the results of this work to those obtained by M. Grafoute et al. with $Q(N_2)$ = 2 sccm [16,17]. The increase in the oxygen flow rate leads to a rapid enrichment of the oxygen content that comes with a strong decrease in the nitrogen content in the films.

## 2. Materials and Methods

Iron oxynitride coatings of about 500 nm were deposited by magnetron sputtering of iron target in Ar-$N_2$-$O_2$ reactive mixtures using a conventional process. Reactive sputtering experiments were performed using Alliance Concept AC450 sputter equipment with a vacuum chamber volume of about 70 L. A base pressure of $10^{-5}$ Pa was obtained with a turbomolecular pump backed with a mechanical one. The working Ar pressure was kept constant at 0.3 Pa usingBrooks mass flow rate controllers and a constant pumping speed $S$ = 10 L s$^{-1}$. A pure (99.5%) metallic iron disc of 50 mm in diameter, which was located at 60 mm from the substrate, was dc sputtered with a constant current density of 100 A m$^{-2}$. The substrates were cleaned with acetone and alcohol before charging in the deposition chamber. Presputtering for 10 min in pure argon was carried out to clean the iron target. Within the conventional process, nitrogen flow rate $Q(N_2)$ value was chosen at 8 sccm. Comparisons are performed sometimes with previous films [16] deposited with gas flow parameters: $Q(N_2)$ = 2 sccm. The oxygen flow rate $Q(O_2)$ was varied between 0 and 1.6 sccm. The Fe-O-N coatings were deposited without external heating. Thus, the deposition temperature was expected to be lower than 323 K.

The thickness of the films deposited on glass and (100) silicon substrates was measured with a Dektak 3030 profilometer. The deposition rates were calculated from the sputtering time.

X-ray diffraction (XRD) patterns were obtained using Phillips X'pert diffractometer with Cu Ka radiation. A wavelength of $\lambda$ = 1.5405 Å was used to obtain the data at a grazing angle of 0.7°. Rutherford backscattering equipment (RBS) was used to estimate the films compositions. RBS measurements were performed with the van de Graaff accelerator using a 2 MeV He$^+$ beam and 2 MeV proton beam. Proton was used because of the increased sensibility for light elements such as nitrogen or oxygen.

Mössbauer spectrometry was also used to observe the different types of Fe environment in these Fe-O-N coatings films. Mössbauer spectra of the films deposited on silicon have been measured at room temperature (300 K) by Conversion Electron Mössbauer Spectrometry (CEMS) in a standard reflection geometry with a constant acceleration signal and a $^{57}$Co source diffused into a rhodium matrix using a He/$CH_4$ gas flow proportional counter. The Mössbauer spectrometer, precisely the Rikon 5 (ORTEC) was calibrated using

$\alpha$-Fe, and the isomer shift values are given relative to that of $\alpha$-Fe at room temperature. The spectra were fitted with the MOSFIT program.

The films' electrical resistivity was deduced from sheet resistance measurements by the four-point probe method using a JANDEL device at room temperature. The conductivity of the films was evaluated between 20 and 170 °C using the Van der Pauw method. The optical transmittance of approximately 500 nm-thick films deposited on glass substrates was studied by UV–visible spectroscopy in the 200–1100 nm range using a Perkin Elmer Lambda 950 spectrophotometer. Finally, the refractive index (n) and the extinction coefficient (k) were deduced from spectroscopy ellipsometry analysis at an incidence angle of 70° in a 0.75–4.5 eV energy range (1700–270 nm), with a step of 0.02 eV.

## 3. Results and Discussion

### 3.1. Chemical Composition

Table 1 presents the concentration of iron, nitrogen, and oxygen in the films. It is important to emphasize that no traces of minor elements have been detected in the films (detection limit about 1 at.%).

**Table 1.** Chemical composition, Fe/(O + N) and chemical formula determined by RBS measurements of the films deposited with $Q(N_2)$ = 8 sccm and varying $Q(O_2)$ from 0 to 1.6 sccm.

| Q(O₂) sccm | Fe (% at.) | N (% at.) | O (% at.) | Q(O₂)/ (Q(O₂) + Q(N₂)) | Fe/(O + N) | Chemical Formula | Compound Type |
|:---:|:---:|:---:|:---:|:---:|:---:|:---:|:---:|
| 0 | 50.9 | 49.1 | 0 | 0 | 1.04 | $Fe_{1.04}N$ | $\gamma''$-FeN |
| 0.4 | 41.2 | 12.9 | 45.9 | 0.05 | 0.70 | $Fe_{2.10}O_{2.34}N_{0.66}$ or $Fe_{2.80}O_{3.12}N_{0.88}$ | Maghemite-like ($\gamma$-$Fe_2O_3$) Magnetite-like ($Fe_3O_4$) |
| 0.6 | 40.7 | 8.8 | 50.5 | 0.07 | 0.69 | $Fe_{2.06}O_{2.55}N_{0.45}$ or $Fe_{2.76}O_{3.41}N_{0.59}$ | |
| 0.8 | 39.9 | 5.5 | 54.6 | 0.09 | 0.66 | $Fe_{1.95}O_{2.69}N_{0.31}$ | |
| 1.0 | 39.4 | 6.2 | 54.4 | 0.11 | 0.65 | $Fe_{1.99}O_{2.73}N_{0.27}$ | Hematite-like ($\alpha$-$Fe_2O_3$) |
| 1.4 | 39.7 | 2.2 | 58.1 | 0.15 | 0.66 | $Fe_{1.98}O_{2.89}N_{0.11}$ | |
| 1.6 | 39.2 | 2.2 | 58.5 | 0.17 | 0.65 | $Fe_{1.94}O_{2.89}N_{0.11}$ | |

For the film deposited without oxygen, the RBS result indicates nearly the same average nitrogen and iron contents, 50%. From the data, the stoichiometry of the sample was calculated to be $Fe_{1.04}N$. Regarding the available literature about the iron nitrides [18,19], the structure of this film can be easily attributed to the iron nitride FeN with ZnS or NaCl structural type. The XRD analysis will give an answer to this assertion.

When the oxygen flow rate $Q(O_2)$ increases from 0.6 to 1.6 sccm, the nitrogen concentration decreases from 8.8 to 2.2% and the oxygen concentration increases from 50 to 58%.

It is interesting to note that the concentration of iron seems to be unaffected by the $Q(O_2)$ increase from 0.6 to 1.6 sccm. Then, from Figure 1, the ratio Fe/(O + N) for films containing oxygen, is ranged from 0.65 to 0.7.

Based on the RBS results, we have calculated the composition of the films as summarized in Table 1. The presence of both elements in these films deposited with $Q(O_2) \geq 0.4$ sccm, i.e., oxygen and nitrogen, confirms that the films are not pure oxide but rather oxidelike oxynitrides. These results of chemical composition clearly indicate the transition from iron nitride to oxidelike oxynitrides films. Regarding the films grown at $Q(O_2)$ = 0.4 sccm ($Fe_{2.1}O_{2.34}N_{0.66}$) and 0.6 sccm ($Fe_{2.06}O_{2.55}N_{0.45}$), we note that they present an overstoichiometry (Table 1) on Fe in comparison to the chemical formula type of $Fe_2(O, N)_3$. In addition, for these two films, a chemical formula type of $Fe_3(O,N)_4$ can be proposed: $Fe_{2.80}O_{3.12}N_{0.88}$ and $Fe_{2.76}O_{3.41}N_{0.59}$. Earlier, research groups [20,21] have shown that it is possible to obtain

intermediate nonstoichiometric $Fe_{3-\delta}O_4$ films between $Fe_3O_4$ and $\gamma$-$Fe_2O_3$ by adjusting the $NO_2$ pressure during deposition by $NO_2$-assisted molecular beam epitaxy. The formula representation of $Fe_3$-$O_4$ is $[Fe^{3+}]_{tet}[Fe_{1-3\delta}{}^{2+},Fe_{1+3\delta}{}^{3+},]_{oct}O_4$, indicating that the $Fe^{2+}$ ions and the vacancies occupy octahedral sites and that the $Fe^{3+}$ ions are distributed evenly over octahedral and tetrahedral sites. These results are an important indication for the possible compound type that may be formed in the films, which will be then accurately investigated by crossing with the results of both X-ray diffraction (XRD) and Conversion Electron Mössbauer spectrometry (CEMS).

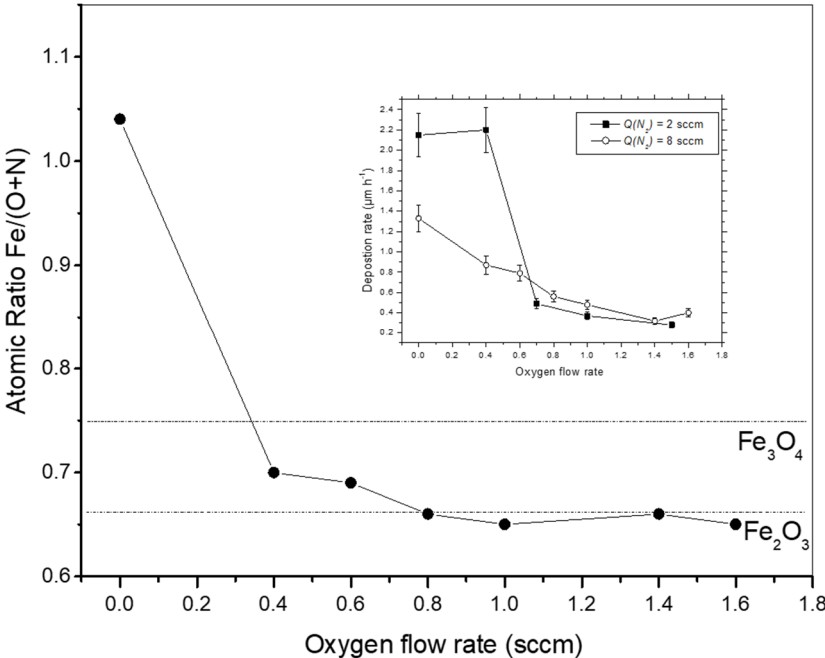

**Figure 1.** Influence of the oxygen flow rate on the Fe/(O + N) atomic ratio chemical composition of films deposited with $Q(N_2)$ = 8 sccm for $Q(O_2)$ varying from 0 to 1.6 sccm. In the insert is the deposition rate versus the oxygen flow rate.

### 3.2. XRD Analysis

Figure 2 presents the X-ray diffractograms of the films. We note a structural change from iron nitride to iron oxide ones.

Thus, for $Q(O_2)$ = 0 sccm, the diffraction peaks correspond to both $\gamma''$-FeN or $\gamma'''$-FeN. As mentioned previously, the nitrogen and iron concentrations obtained by RBS measurements are nearly equal to 50 at.% each. This result is in agreement with the XRD analysis. The structure of the $\gamma''$-FeN was determined to be ZnS-type, whereas the $\gamma'''$-FeN was Nacl-type [18,22]. In the ZnS-type structure, the nitrogen atoms are in the tetrahedral sites of the fcc iron lattice. In the NaCl-type structure, all the octahedral sites are filled by the nitrogen atoms. Many authors [23] have reported in the literature that the lattice parameter of the $\gamma''$-FeN lies between 0.428 nm and 0.433 nm. In the case of the present work, the film is well-crystallized and the lattice parameter calculated is 0.431 nm. Thus, based on this lattice parameter, this film is attributed to the $\gamma''$-FeN with the ZnS structure type. It is well-established that the lattice constant of the $\gamma'''$-FeN is around 0.45 nm. Taking into account of the presence of probable texture in the film, the diffraction intensity cannot be used to differentiate the two structures. However, it is important to emphasize that the lattice constants of the Fe-N system ($\gamma''$: 0.431nm and $\gamma'''$: 0.45nm) are not so close. I. Jouanny et al. [24] have reported that the $\gamma'''$-FeN phase crystallizes in the ZnS-type structure. In their study, the $\gamma'''$ may be a disordered and nonstoichiometric form of the $\gamma''$ structure. Let us note that no trace of texture is evidenced in their film and the lattice parameter is around 0.455 nm.

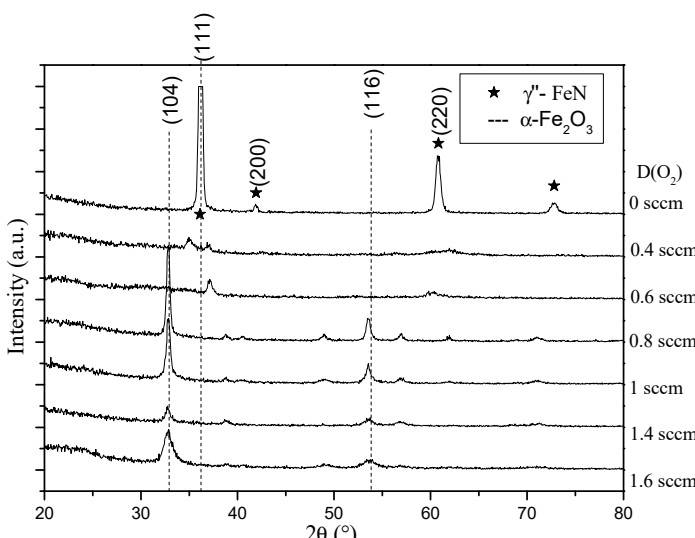

**Figure 2.** Influence of the oxygen flow rate on the X-ray diffractograms of films deposited with $Q(N_2)$ = 8 sccm for $Q(O_2)$ varying from 0 to 1.6 sccm (JCPDS 50-1087 and 89-0596).

Regarding the film grown at $Q(O_2)$ = 0.4 sccm or 0.6 sccm, we note that their structure (Figure 2) changes and is poorly crystallized. However, the diffraction peak detected after XRD analysis corresponds to common reflections of maghemite ($\gamma$-Fe$_2$O$_3$) and magnetite (Fe$_3$O$_4$). Thus, it is difficult to differentiate between the two oxides from their X-ray diffraction pattern (Figure 3b).

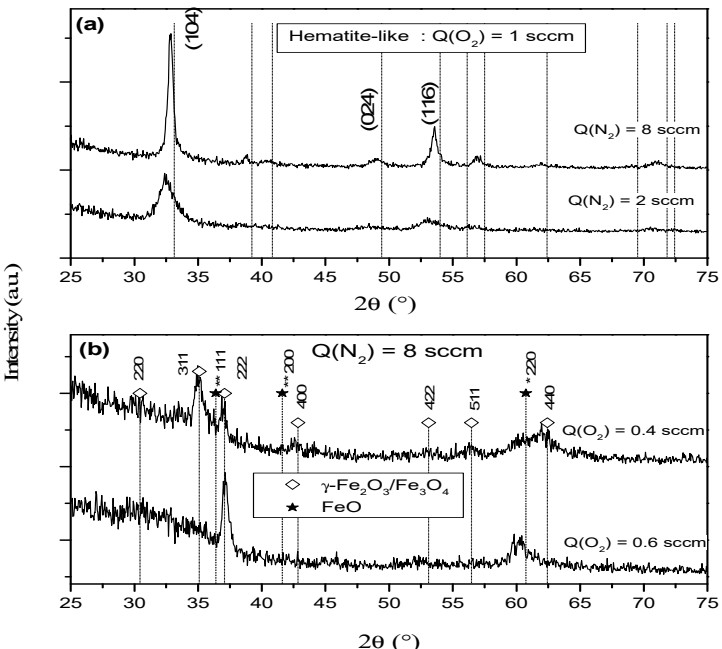

**Figure 3.** Comparison of (**a**) the X-ray diffractograms of hematite-like films deposited with $Q(N_2)$ = 2 sccm and $Q(N_2)$ = 8 sccm for $Q(O_2)$ = 1 sccm and (**b**) X-ray diffractograms of $\gamma$-Fe$_2$O$_3$/Fe$_3$O$_4$ deposited with $Q(O_2)$ = 0.4 sccm and $Q(O_2)$ = 0.6 sccm for $Q(N_2)$ = 8 sccm (JCPDS89-0596).

It is well-known that magnetite has an inverse spinel structure with Fe$^{3+}$ ions distributed randomly between octahedral and tetrahedral sites, and Fe$^{2+}$ ions in octahedral sites. Maghemite has a spinel structure that is similar to that of magnetite but with Fe$^{2+}$ ions replaced by vacancies in sublattice. Then, several solutions can be proposed, for instance, the possibility of having an intermediate compound [20,21] Fe$_{3-\delta}$O$_4$ between Fe$_3$O$_4$ and

$\gamma$-Fe$_2$O$_3$, or a mixture of oxide [25,26] $\gamma$-Fe$_2$O$_3$/Fe$_3$O$_4$ or $\gamma$-Fe$_2$O$_3$/FeO, $\gamma$-Fe$_2$O$_3$/FeON, or $\gamma$-Fe$_2$O$_3$ can be assumed. As previously mentioned, these samples or oxidelike present an over- or understoichiometric on Fe in comparison to the chemical formula type of Fe$_2$(O, N)$_3$ or Fe$_3$(O, N)$_4$, respectively. Let us remember that the oxynitride film Fe$_{1.06}$O$_{0.35}$N$_{0.65}$ [15,16] prepared with $Q$(N$_2$) = 2 sccm and $Q$(O$_2$) = 0.4 sccm, crystallizes in a face-centered-cubic structure type NaCl with a lattice parameter estimated at 0.452 nm. Contrary to this film Fe$_{1.06}$O$_{0.35}$N$_{0.65}$, the chemical formula obtained by RBS is Fe$_{2.10}$O$_{2.34}$N$_{0.66}$ or Fe$_{2.80}$O$_{3.12}$N$_{0.88}$ for this film prepared with $Q$(N$_2$) = 8 sccm and $Q$(O$_2$) = 0.4 sccm. This reveals the difference between both samples ($Q$(N$_2$) = 2 sccm and $Q$(N$_2$) = 8 sccm) prepared with the same oxygen flow rate $Q$(O$_2$) = 0.4 sccm. In addition, the deposition rate for both series of films deposited with $Q$(N$_2$) = 2 sccm and $Q$(N$_2$) = 8 sccm, respectively, is shown in the insert in Figure 1. It is important to note that the initial deposition rate is around 2.1 $\mu$m.h$^{-1}$ and 1.3 $\mu$m.h$^{-1}$ for the iron nitrides $\varepsilon$-Fe$_{2.2}$N and $\gamma''$FeN, respectively. We also observe a progressive decrease in the deposition rate for samples prepared with $Q$(N$_2$) = 8 sccm. For the samples deposited with $Q$(N$_2$) = 2 sccm, increasing the oxygen content up to 0.4 sccm, the deposition rate remains constant and the value is close to that of the iron nitride $\varepsilon$-Fe$_{2.2}$N. On this basis, it is clear that the absence of oxynitride film type Fe(O, N) in the case of this work can be justified by the lower deposition rate of this process and by the absence of a rich phase on iron phase such as $\varepsilon$-Fe$_{2.2}$N. This result is in total agreement with the chemical composition and the X-ray diffraction results.

Finally, when $Q$(O$_2$) exceeds 0.6 sccm, the X-ray patterns confirm the oxide phase $\alpha$-Fe$_2$O$_3$ formation with a progressive change on the preferential growth of the (104) peak [13,26]. Similar behavior [16] has been observed when preparing FeON films with $Q$(N$_2$) = 2 sccm. Regarding the evolution of the oxygen flow rate, there is an increase in the full width half maximum (FWHM) of the diffraction peaks. This is the evidence for the reduction in the grain size. In order to understand the similarities and the differences between these series ($Q$(N$_2$) = 8 sccm) and the series ($Q$(N$_2$) = 2 sccm) reported in the literature [15,16], a detailed analysis was carried out by comparing the samples of both series, with similar oxygen content in Figure 3. The diffractograms (Figure 3a) reveal that for the same oxygen flow rate Q(O$_2$) = 1 sccm, the sample deposited with high nitrogen flow rate is rather well-crystallized. To understand the Q(N$_2$) effect in the films structure and morphology, the coherent diffraction domain size was calculated considering the (104) plane (Figure 3a) and using Scherrer equation. From the equation, the crystalline domain size was found as approximately 5 nm and 18 nm for the films Q(N$_2$) = 2 sccm and Q(N$_2$) = 8 sccm, respectively.

These differences should result mainly from the presence of a higher quantity of nitrogen in the deposition chamber. It is thus clear that a high concentration of nitrogen plays a fundamental role in the crystallization of the hematite. Indeed, the crystallization of hematite is improved when the ratio (Q(O$_2$)/(Q(O$_2$) + Q(N$_2$))) decreases from 0.33 (when Q(N$_2$) = 2 sccm) to 0.11 ((when Q(N$_2$) = 8 sccm). This behavior could be the result of the catalytic effect of nitrogen on the crystallization of the hematite. In addition, such results have been observed by S. Venkataraj et al. [2] on the growth of zirconium oxynitride films prepared by reactive direct current magnetron sputtering. These authors show that the incorporation of nitrogen in the films improves the crystalline quality.

### 3.3. $^{57}$Fe Mössbauer Spectrometry

Conversion Electron Mössbauer Spectrometry (CEMS) spectra for the films deposited with a nitrogen flow rate of 8 sccm are illustrated in Figure 4 at room temperature 300 K.

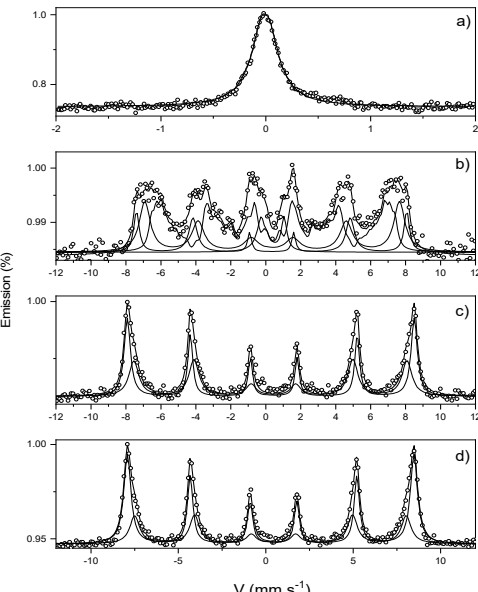

**Figure 4.** Mössbauer spectra of films prepared with a nitrogen flow rate of 8 sccm at room temperature 300 K. (**a**) $Q(O_2) = 0$ sccm, (**b**) $Q(O_2) = 0.4$ sccm, (**c**) $Q(O_2) = 0.8$ sccm, and (**d**) $Q(O_2) = 1$ sccm.

The film (Figure 4a) deposited without oxygen shows a paramagnetic spectrum at room temperature, which is almost single phase of ZnS-type nitride. The fitting is well-achieved by considering one singlet characterized by an isomer shift value of 0.09 mm/s. Based on the Mössbauer data reported in the literature by Schaaf et al. [27] or Hinomura et al. [28], this singlet can be attributed to $\gamma''$-FeN in the ZnS structure all having the nearest nitrogen neighbor sites occupied. Compared with the studies of different research groups [18,29], it is the first time that the monophased $\gamma''$-FeN has been evidenced.

When Q(O₂) is fixed at 0.4 sccm, the spectrum (Figure 4b) obtained is broadened with a magnetic behavior at room temperature that cannot be satisfactorily fitted with the combination of magnetite ($Fe_3O_4$) and maghemite ($\gamma$-$Fe_2O_3$) [30]. This result confirms that this film is poorly crystallized and is consistent with the X-ray diffraction patterns. The best fit is achieved by using a paramagnetic doublet having an isomer shift (quadrupolar splitting) value of 0.40(2) mm s$^{-1}$ (1.19(2) mm s$^{-1}$), two sextets, and a distribution of hyperfine fields linearly correlated with a constant isomer shift. Thus, the doublet with isomer shift corresponds to ferric ion $Fe^{3+}$. Similar results with isomer shift of 0.46 mm s$^{-1}$ and quadrupolar splitting of 1.19 mm s$^{-1}$ have been obtained in Mössbauer spectra of $\gamma$-$Fe_2O_3$ nanoparticles. The values of the hyperfine parameters obtained for the sextets are as follows: IS = 0.35(2) mm/s ($B_{hf}$ = 47.7(2) T) and 0.39(2) mm/s (45.0(2) T), respectively. The two sextets are unambiguous to $Fe^{3+}$ sites characteristics of the $\gamma$-$Fe_2O_3$ phase. F. C. Voogt et al. [13] have produced magnetite-like oxynitride films with composition $Fe_{3+\delta}O_{4-y}N_y$ at low NO₂ fluxes (0.08 < $\delta$ < 0.16 and 0.19 < y < 0.36) by NO₂-assisted molecular beam epitaxy. In their paper, the Mössbauer spectra of the films, which are typical for the magnetite, were described with three components ($Fe^{3+}$, $Fe^{2.5+}$, and Fe-N) and the nitrogen atoms occupied substitutional sites of the oxygen anion sublattice by forming $Fe_{3+\delta}(O, N)_4$. Regarding the isomer shift, which provides valuable information of the oxidation state, one notes that the contribution of the $Fe^{2.5+}$ sites present in the magnetite phase with isomer shift of 0.66 mm s$^{-1}$ is absent in the hyperfine parameters of this sample since there is clear indication of the absence of the magnetite phase or the intermediate compound $Fe_{3-\delta}O_4$ phase between $Fe_3O_4$ and $\gamma$-$Fe_2O_3$. Due to the presence of 12.9 at.% of nitrogen (Table 1), the last component corresponding to the distribution ($B_{hf}$ = 45(2) − 15(2) T) is attributed to iron having at least one nitrogen as first neighbor. No trace of the paramagnetic wüstite FeO with isomer shift of 0.93(2) mm s$^{-1}$ and quadrupole splitting of 1.11(2) mm s$^{-1}$ has been detected in the film. Indeed, the Mössbauer results in-

dicate that the major contribution of this film is rather due to the maghemite-like oxynitride phase and that is consistent with the chemical formula obtained by RBS ($Fe_{2.10}O_{2.34}N_{0.66}$). This result confirms the disordered phase and indicates the correlation with the X-ray diffraction patterns and the RBS results.

Finally, for films prepared with higher oxygen flow rate, the Mössbauer spectra consist of a magnetic component with broadened and symmetrical lines that has to be described by at least two sextets. They correspond to two different Fe sites ($Fe^{3+}$ and Fe-N). Analysis of the spectrum shown in Figure 4c,d yields $B_{hf}$ = 50.6(2) T, IS = 0.37(2) mm s$^{-1}$ and $B_{hf}$ = 47.9(2) T, IS = 0.36(2) mm s$^{-1}$ for the two sextets, respectively. The first subspectrum is interpreted as arising from $Fe^{3+}$ ions and corresponds to the $\alpha$-$Fe_2O_3$; in such a case, it is necessary to include the quadrupolar shift, which has a typical value of $-0.15(2)$ mm s$^{-1}$. The second subspectrum is interpreted as arising from Fe ions that have at least one nitrogen nearest neighbor. Contrary to the previous paper [17], this result confirms the effective crystallization of the film prepared with $Q(N_2)$ = 8 sccm and $Q(O_2)$ = 1 sccm. Thus, as the sample prepared with 2 sccm of nitrogen, we can conclude that these films are hematite-like.

### 3.4. Electrical Properties

The effect of the oxygen flow rate on the electrical resistivity of the films is displayed in Figure 5.

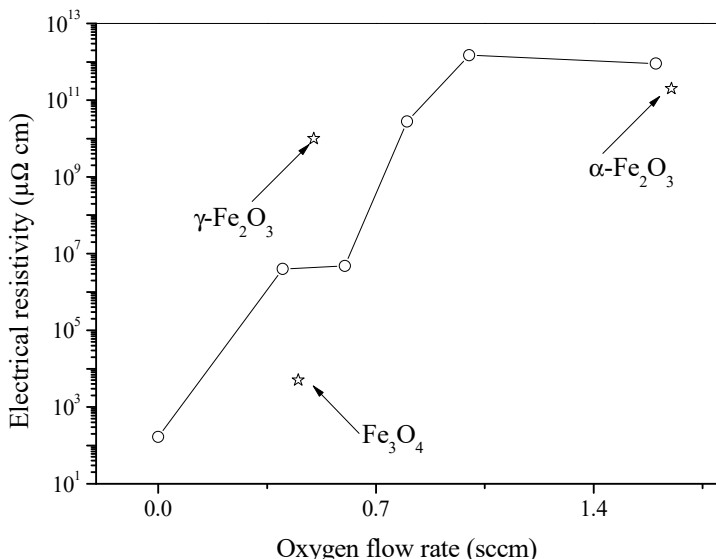

**Figure 5.** Influence of the oxygen flow rate on the electrical resistivity of Fe-O-N coatings deposited with 8 sccm. The star symbol is related to the electrical resistivity of amorphous iron oxide film deposited without nitrogen.

The nitride coating exhibits an electrical resistivity lower than 200 μΩ cm, which is consistent with the metallic character of iron nitride [31]. The behavior of the resistivity in nitrides is a very complex subject, depending not only on composition but also on parameter such as crystalline structure [32,33]. Thus, comparing this value (200 μΩ cm) to the value reported by I. Jouanny et al. [24] (270 ± 14 μΩ) in the $\gamma''$-FeN structure type, we note the correlation between the resistivity and the structure determined in this work.

Introduction of oxygen with flow rate of 0.4 sccm induces a strong increase in the films' electrical resistivity. From 0.4 sccm to 0.6 sccm, the electrical resistivity remains almost constant and the values are around $4.01 \times 10^6$ μΩ cm and $4.77 \times 10^6$ μΩ cm, respectively. The structure of these poorly crystallized films, correspond to maghemite or magnetite. It is well-known that the resistivity of the magnetite is $5 \times 10^3$ μΩ cm [34]. B. Mauvernay et al. [35] has shown that the resistivity versus the partial pressure of iron oxide ($Fe_3O_4$ + FeO) ranges from $2.8 \times 10^5$ to $12.3 \times 10^5$ μΩ cm. Note that the XRD results

indicate the absence of the FeO in the samples (0.4 sccm and 0.6 sccm). Thus, this high value of electrical resistivity ($4.01 \times 10^6$ μΩ cm or $4.77 \times 10^6$ μΩ cm) is an important indication for the absence of the magnetite in the sample. Based on the electrical resistivity results and the literature [34,35], we can say that the film deposited with 0.4 sccm and 0.6 sccm may be closed to maghemite.

With the further increase in $Q(O_2)$, the films crystallize in the $\alpha$-Fe$_2$O$_3$ structure and the films exhibit an electrical resistivity higher than $10^{10}$ μΩ cm, which is consistent with the value measured on oxide films [35] deposited without nitrogen ($2 \times 10^{11}$ μΩ cm). One concludes that the Fe-O-N electrical properties are strongly correlated with the film's composition and structure. Similar results have been reported in literature; for instance, Chappé et al. [36] reported that the increase in oxygen content and the increase in ionic bonding character are responsible for the smooth increase in electrical resistivity in the transition between nitride and oxide sputtering regimes for the TiOxNy system.

In Figure 6, we display the electrical conductivity versus the inverse temperature of the disordered films prepared with these parameters:

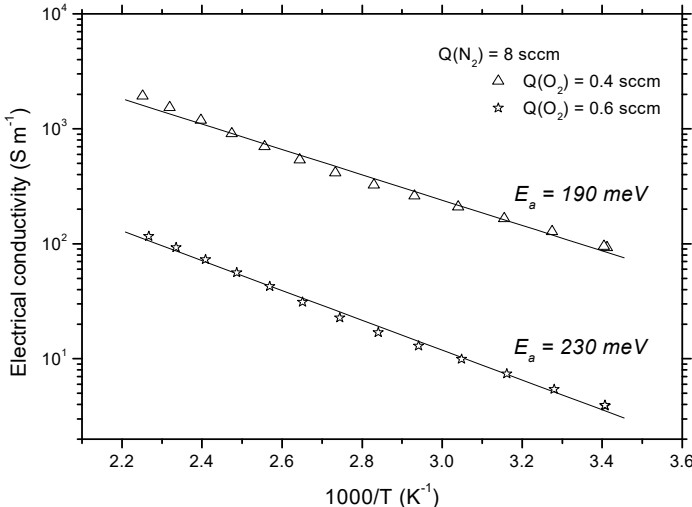

**Figure 6.** The electrical conductivity versus the inverse temperature of the oxynitride films deposited at $Q(N_2)$ = 8 sccm with Q(O$_2$) = 0.4 and 0.6 sccm.

We can observe in Figure 6 an increase in the electrical conductivity when the temperature increases. This attributes a semiconductor behavior for these films. It is well-known that $\alpha$-Fe$_2$O$_3$ (hematite) is a n-type semiconductor [37]. However, according to Morikawa et al. and Ogawa et al. [37,38], it can change to p-type conduction induced by N-doping in $\alpha$-Fe$_2$O$_3$. Since we did not perform any hall effect measurements in this study, we can assume that our hematite-like films are of p-type.

In graph 6, the activation energies are calculated assuming a linear behavior in an Arrhenius plot. The activation energy of the disordered films is close and the values are important around 190 meV and 230 meV, respectively. Indeed, during our previous study [16] the measured activation energy of the well-crystallized film deposited with $Q(N_2)$ = 2 sccm and $Q(O_2)$ = 0.4 sccm was found to be 35 meV. In this present study, we increased the nitride flow rate ($Q(N_2)$ = 8 sccm), and the disordered film deposited with the same $Q(O_2)$ = 0.4 presented an activation energy of 190 meV. Therefore, we can assume that the high values of activation energy shown in Figure 6 are due to structural defects in the disordered films. Furthermore, the activation energy of conductivity increases when the oxygen flow rate or partial pressure increases. This may be due to the resistive nature of the O-rich films and to the effects of grain boundary scattering in the films. This evolution has been also observed by Miller et al. [39]. This resistive nature of O-rich films is also responsible for the decrease in the conductivity when the oxygen flow rate increases.

These results confirm the correlation between the structure of both disordered films and the catalytic effect of the nitrogen. Thus, coatings with $Fe_2(O, N)_3$ oxide-type structure exhibit very high electrical resistivity associated with a semiconducting behavior [39].

### 3.5. Optical Properties

Optical properties of Fe-O-N coatings have been investigated by UV–visible spectroscopy in the 300–1100 nm range. The transmittances of the coatings are plotted in Figure 7 as a function of the wavelength.

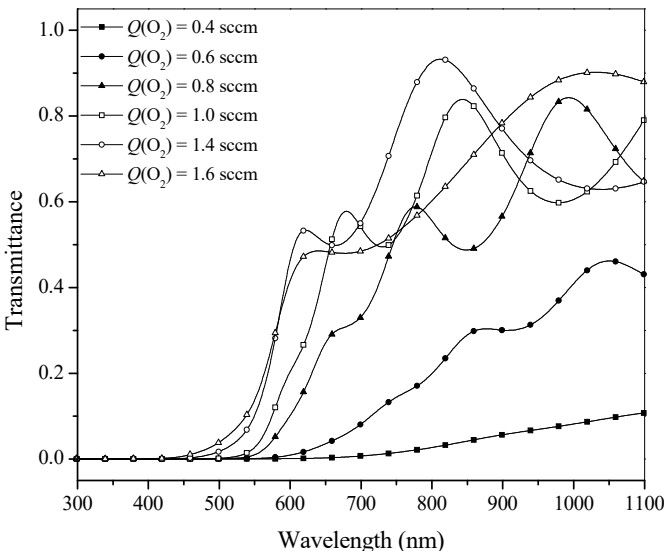

**Figure 7.** Influence of the oxygen flow rate on the transmittance of Fe-O-N films ($Q(N_2)$ = 8 sccm).

It is important to note the progressive change of the transmittance when the oxygen flow rate increases. Almost all the films are opaque in the UV range and in the visible range up to 450 nm. However, we note a small shift of the threshold of absorption toward smaller wavelengths, going with the increase in the oxygen flow rate. In addition, the films are slightly transparent for samples deposited with oxygen flow rate higher than 0.6 sccm, as previously reported [40]. In this plot, it is possible to observe that the last two samples present different optical behaviors, in total agreement with the composition and the crystalline results. Lei Zhang et al. [41] have shown that the absorption edge of the bulk $\gamma$-$Fe_2O_3$ reference is around 560 nm-wavelength. It is important to note that this absorption edge value is consistent with the value of this work (570 nm) for the disordered samples (0.4 sccm and 0.6 sccm) and confirms the maghemite phase of these films. One more time, the transmittance measurements confirm the difference between nitride, intermediate oxide-type (maghemite-like), and oxide-type (hematite-like) coatings.

From Figure 7, we have determined the energy gap between the highest filled valence band and the lowest total empty conduction band of the samples. Figure 8 shows the optical direct and indirect gap for representative samples as a function of the oxygen flow.

Let us remember that Yoko et al. [42] have reported in the literature the presence of two types of transitions in the $\alpha$-$Fe_2O_3$. Thus, considering the direct transition, there is a progressive increase from 1.6 eV to 2.5 eV. Regarding the increase in the direct energy gap for the samples, the main reason should result from the decrease in the grain size versus the oxygen flow rates, as previously indicated. On the other hand, considering the indirect transition, it seems clear that the optical gap is unaffected by the oxygen flow higher than 0.6 sccm. Thus, the value of the indirect energy gap of the hematite-like films is around 1.7 eV. This result is similar to those of the literature [39,40].

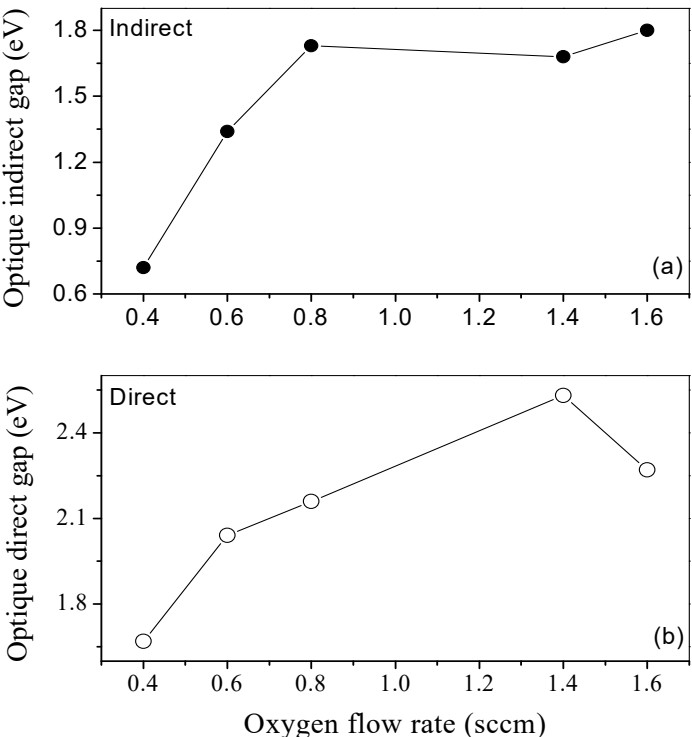

**Figure 8.** (**a**) Optical direct and (**b**) indirect gap for representative samples deposited with $Q(N_2)$ = 8 sccm as a function of the oxygen flow.

A close look into Figure 8 shows that the reduction of oxygen partial pressure in the process leads to the decrease in the band gap. The energy gap of the magnetite film reported in the literature is 0.1 eV [11] or 0.3 eV [43], whereas it is higher for maghemite thin film. Optical properties of maghemite thin film were not studied in detail in the literature but the expected value of its direct optical band gap is larger than 2.46 eV [44]. Mirza et al. [45] obtained an optical bandgap of 2.3 eV for a direct transition. In this study, the direct optical bandgaps of the disordered films are 1.7 eV and 2 eV. Thus, based on the direct optical band gap, and since our direct optical bandgap values are close to those of Mirza et al. [45], we can assume that the disorder films could be maghemite-like films.

This optical band gap reduction observed between hematite-like film and maghemite-like film is correlated with the increase in the ratio N/O. It is important to note that this behavior has been also observed on transition metal oxynitrides coatings TiON [6], NbON [46], CrON [47], TiON [48].

The refractive index (*n*) and the extinction coefficient (*k*) have been deduced from spectroscopic ellipsometry analysis at an incidence angle of 70° in an energy range 0.75–4.5 eV (270–1700 nm), with a step of 0.02 eV. The results are displayed in Figure 9.

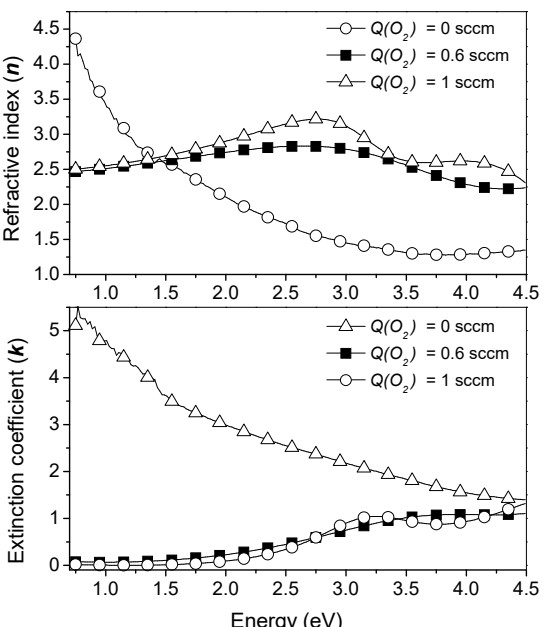

**Figure 9.** Effect of the oxygen flow rate on the refractive index and extinction coefficient of Fe-O-N films deposited with $Q(N_2)$ = 8 sccm.

Indeed, for $Q(O_2)$ = 0.6 sccm and 1 sccm, *n* and *k* of Fe-O-N coatings seem to be unaffected by the variation in $Q(O_2)$, since their evolution versus photon energy is very close. However, the nitride film deposited without oxygen shows the values of n and k different to those of the oxides. So, in accordance with XRD and RBS analysis, n and k evolution reveals the formation of nitride- and oxides-likes. Moreover, at low photon energy, *k* is very close to 0 for the films deposited with $Q(O_2)$ = 0.6 sccm and 1 sccm. This indicates that these films exhibit dielectric properties observed in oxide films [49].

## 4. Conclusions

Fe-O-N films have been synthesized by reactive magnetron sputtering of an iron target at high nitrogen partial pressure of 1.11 Pa with different oxygen flow rates and compared with previous Fe-O-N films obtained at a low nitrogen partial pressure of 0.25 Pa. At low nitrogen partial pressure of 0.25 Pa, and before addition of oxygen flow rate, the films exhibited a $\varepsilon$-fe$_2$N structure with an iron to nitrogen atomic ratio close to 2.2 [16]. In this study, the sputtering process implemented with higher nitrogen partial pressure of 1.11 Pa excluded the presence of the $\varepsilon$-Fe$_2$N structure and promoted the emergence of a single phase of $\gamma''$-FeN with the ZnS-type structure. However, when the oxygen was introduced in the sputtering process with $Q(O_2)$ > 0.6 sccm, the films exhibited a Fe/(O + N) chemical ratio close to 0.66 with a well-crystallized structure and their optical and electrical properties were consistent with those of an $\alpha$-Fe$_2$O$_3$ structure. Comparatively to a previous study carried out at low nitrogen partial pressure (0.25 Pa), this behavior of films prepared at higher nitrogen partial pressure (1.11 Pa) could result from a catalytic effect of nitrogen on the crystallization of the hematite structure. At intermediate oxygen flow rates, 0.4 and 0.6 sccm, the deposited films were rather poorly crystallized, according to the XRD and Mössbauer results. The XRD results indicate that films deposited with 0.4 and 0.6 sccm can be attributed to a maghemite or magnetite phase or a mixture of these two phases. The results of XRD and composition are not sensible enough to conclusively determine the structural phase obtained. However, the electrical and optical results indicated that these films deposited with 0.4 and 0.6 sccm are very close to maghemite. The simulations of Mössbauer spectra do not reveal the presence of the intermediate ions denoted by the average state Fe$^{2.5+}$ in these samples. Therefore, it is unlikely that these films form the maghemite-like oxynitride phase and the chemical formula obtained by RBS is $\gamma$-

Fe2.10O2.34N0.66. The Mössbauer analysis of these samples, especially at low temperature, would be necessary to try to understand more precisely the structure of these two samples.

Due to the occurrence of high nitrogen partial pressure, the reactive sputtering processes can be accessed for the first time to the monophased $\gamma''$-FeN. Further, this process allows more easily, by a catalytic effect of nitrogen, the synthesis of hematite crystallized films with better optical and electrical properties.

**Author Contributions:** Conceptualization, C.R., J.F.P., M.G., and C.P.; methodology, C.R., J.F.P., M.G., and C.P.; validation, C.R., J.F.P., M.G., and C.P.; investigation, C.R., J.F.P., M.G., C.P., and K.B.J.-I.N.; resources, C.R., J.F.P., M.G., and C.P.; data curation, M.G. and C.R.; writing—original draft preparation, M.G. and C.P.; writing—review and editing, C.R., J.F.P., M.G., C.P., and K.B.J.-I.N.; visualization, C.R.; supervision, C.R., J.F.P., and M.G.; project administration, C.R. and J.F.P.; funding acquisition, C.R. and J.F.P. All authors have read and agreed to the published version of the manuscript.

**Funding:** This work has been financially supported by the European project "HARDECOAT": NMP3-CT-2003-505948, ITSFC, Pays Montbéliard Agglomération, Région Franche-Comté, DRIRE and FEDER.

**Institutional Review Board Statement:** Not applicable.

**Informed Consent Statement:** Not applicable.

**Data Availability Statement:** All data were presented in this manuscript.

**Conflicts of Interest:** The authors declare no conflict of interest.

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
