# Peer review of "Influence of Oxygen Flow Rate on the Properties of FeOXNY Films Obtained by Magnetron Sputtering at High Nitrogen Pressure"

_coatings, doi:10.3390/coatings12081050_

Round 1

Reviewer 1 Report

The manuscript corresponds to the topics of the Coatings. The manuscript has written at a high level. Many experimental methods were used. Before publishing, it is necessary to take into account the following comments:

1. What is the interest in the study of oxynitride and nitride films of iron? Practical applications of these films?

2. I suggest that the authors should compare the characteristics of oxynitride and nitride films of iron obtained by various PVD methods in the form of a table.

3. What is "DRX analysis"? There's probably a typo here.

4. Pay attention to ln.161, 162 and 173.

5. What is the thickness of the films?

6. What type of conductivity of films with semiconductor characteristics is?

7. What defect or impurity does cause such values of activation energies? Why does the activation energy of conductivity depend on oxygen pressure?

8. Present and describe the temperature dependences of resistivity for oxynitride and nitride films of iron.

Author Response

Responses to reviewer comments are given in the attached file

Reviewer 2 Report

The authors of manuscript entitled " Influence of oxygen flow rate on the properties of FeOxNy films obtained by magnetron sputtering at high nitrogen pressure", proposed effect of oxygen content on crystal structure of iron oxide by magnetron sputtering.

In this work, authors synthesize different crystal structures of Fe-O-N thin films under different oxygen concentrations, such asγ’’-FeN, α-Fe2O3, and γ-Fe2O3/Fe3O4. And their electrical and optical properties are studied. Optical direct and indirect gaps have been also obtained. It is interesting. I would like to suggest that authors revise the manuscript according to a few questions as follow.

1.     Figure 4c is not mentioned in the text.

2.     There's no any literature from the last ten years (2011-2022). Why?

There is lack of TEM (and Selected Area Electron Diffraction) or STM images which can characterize atomic structure directly.

Round 2

Reviewer 1 Report

The authors took into account the reviewer's comments. The manuscript can be accepted for publication.